# Effects of Methyl Sulfonyl Methane and Selenium Yeast on Fatty Liver Syndrome in Laying Hens and Their Biological Mechanisms

**DOI:** 10.3390/ani13152466

**Published:** 2023-07-30

**Authors:** Huanbin Wang, Lingfeng Wang, Changyu Tian, Shahid Ali Rajput, Desheng Qi

**Affiliations:** 1Department of Animal Nutrition and Feed Science, College of Animal Science and Technology, Huazhong Agricultural University, Wuhan 430070, China; whbin@webmail.hzau.edu.cn (H.W.); wanglingfeng0@webmail.hzau.edu.cn (L.W.); tianchangyu@webmail.hzau.edu.cn (C.T.); 2Faculty of Veterinary and Animal Science, Muhammad Nawaz Shareef University of Agriculture, Multan 60000, Pakistan; shahid.ali@mnsuam.edu.pk

**Keywords:** laying hen, methyl sulfonyl methane, selenium yeast, fatty liver syndrome, antioxidant capability, RNA-seq

## Abstract

**Simple Summary:**

Fatty liver syndrome (FLS) in laying hens has brought serious economic losses to the poultry industry. Therefore, research on feed additives to alleviate FLS is necessary. This study investigated the effects of methyl sulfonyl methane (MSM) and selenium yeast (Se-Y) in the diet of laying hens on production performance, liver steatosis, and antioxidant index and explored the biological mechanisms of MSM and Se-Y to alleviate FLS via RNA-seq technology. The results provide a reference for the prevention and treatment of FLS in laying hens during the late peak laying period.

**Abstract:**

The purpose of this study was to explore the effects of MSM and Se-Y on FLS in laying hens during the late peak laying period and the underlying biological mechanisms. Therefore 240 55-week-old Jing-fen No. 6 laying hens were randomly divided into five groups, with eight replicates in each group and six laying hens in each replicate. The hens were fed a basal diet (Control) and diets supplemented with 350 and 700 mg/kg MSM and 25 and 50 mg/kg Se-Y, respectively, for four weeks. The results showed that MSM and Se-Y had no significant effects on the performance of laying hens. With the increasing dosage of MSM and Se-Y, the symptoms of liver steatosis in laying hens were reduced, and MSM and Se-Y could significantly reduce the content of malondialdehyde (MDA) in serum and liver (*p* < 0.05) and increase the contents of total superoxide dismutase (T-SOD) and glutathione peroxidase (GPX) in serum and liver (*p* < 0.05). The RNA-seq results showed that 700 mg/kg MSM significantly downregulated the expression levels of the ATP5I, ATP5G1, CYCS, and UQCRQ genes in the liver, and 50 mg/kg Se-Y significantly downregulated the expression levels of MAPK10, SRC, BMP2, and FGF9 genes in the liver. In conclusion, dietary supplementation with MSM and Se-Y can effectively reduce the FLS of laying hens in the late peak laying period and increase their antioxidant capacity. The underlying biological mechanism may be related to the downregulation of genes involved in liver oxidative phosphorylation and inflammation-related pathways.

## 1. Introduction

Fatty liver syndrome (FLS) in laying hens is a complex metabolic disease that mainly occurs in laying hens with a high egg production rate and is manifested by liver steatosis, obesity, and decreased egg production [1,2]. Because the symptoms cannot easily be detected and sick hens cannot be treated in time, FLS has caused serious economic losses to the poultry industry [1,2,3]. Oxidative stress injury can increase the level of lipid peroxidation, change the morphology of mitochondria, and damage the biological function of mitochondria, which is an important factor leading to fatty liver in laying hens [4]. In addition, the occurrence of fatty liver is also related to liver inflammation [5]. Long-term inflammation of the liver will cause slow liver damage; if not treated promptly, steatosis will occur and eventually evolve into fatty liver [6]. Recently, FLS has become a bottleneck in the development of the laying hen breeding industry. Adding effective feed additives to the diet to improve the antioxidant levels of laying hens and alleviate their inflammatory symptoms is an important measure to alleviate FLS.

Methyl sulfonyl methane (MSM) is a natural sulfur-containing compound containing dimethyl groups and can reduce oxidative stress [7]. Previous studies have shown that MSM can act as a free radical scavenger in vivo and have the function of scavenging reactive oxygen species and inhibiting their production [8]. In addition, MSM can increase the antioxidant level in tissues, reduce the MDA content in tissues, and protect the liver [7]. At present, the effect of MSM on FLS in laying hens is still largely unclear.

Selenium yeast (Se-Y) is an organic selenium source developed from yeast [9]. Compared with inorganic selenium, the absorption rate of organic selenium by animals is higher [9]. Selenium has anti-inflammatory and antioxidant functions [10], and selenium deficiency can cause the accumulation of reactive oxygen species in organisms and eventually lead to inflammation. Selenium supplementation can effectively block the mitogen-activated protein kinase (MAPK) pathway and reduce inflammation [11,12]. Adding selenium to feed can also effectively reduce the damage caused by inflammation. For example, in mice, Se-Y can inhibit neuroinflammation caused by inorganic aluminum, including the disorder of inflammatory factor expression and the imbalance of the redox system [13]. In addition, selenium is also a commonly used antioxidant in poultry breeding and is mainly used to improve the production performance of poultry and for the production of selenium-rich eggs [7,14]. At present, the effects of Se-Y on FLS in laying hens are still unclear.

To study the effects of MSM and Se-Y on FLS in laying hens at the late peak laying period, in this experiment, 240 55-week-old laying hens were used, investigating the parameters production performance, egg quality, liver steatosis, and antioxidant index of laying hens. In addition, RNA-seq and q-PCR techniques were applied to explore the biological mechanisms of MSM and Se-Y underlying improvement in the liver lipid degeneration of laying hens, providing a reference for the prevention and treatment of FLS in laying hens during the late peak laying period.

## 2. Materials and Methods

### 2.1. Laying Hens, Treatments, and Sample Collection

A total of 240 55-week-old Jing-fen No.6 laying hens were randomly divided into five groups, and each group was divided into eight replicates containing six hens: control group (Fed with a basic diet), MSM experimental group 1 (Fed with 350 mg/kg MSM), MSM experimental group 2 (Fed with 700 mg/kg MSM), Se-Y experimental group 1 (Fed with 25 mg/kg Se-Y), and Se-Y experimental group 2 (Fed with 50 mg/kg Se-Y). Laying hens are raised in the animal metabolism room of Huazhong Agricultural University, Wuhan, China. The basal diet is shown in Table 1 [15]. MSM and Se-Y were added to the powdered basal diet in powder form, respectively, and they were mixed step by step (Additives were mixed with a small amount of basal diet, and then the same amount of basal diet was added for mixing). Repeat this step until the diet is completely mixed. The temperature and photoperiod of the henhouse were the same as those of Wang et al. (2023) [16]. The MSM (Purity ≥ 99.0%) was purchased from Shanghai McLean Biochemical Technology Co., Ltd. (Shanghai, China) and added to the basic diet at 350 and 700 mg/kg. The Se-Y (selenium content: 2000 mg/kg) was purchased from Angel Yeast Co., Ltd. (Wuhan, China) and added to the base diet at doses of 25 and 50 mg/kg. The experiment lasted for four weeks. The production performance of the fourth week was recorded, and the eggs of the fourth week were collected for egg quality analysis. Six laying hens in each group were selected to collect blood from the sub-wing vein and centrifuged at 3000 rpm for 10 min at 4 °C to separate the serum [17]. Killed the laying hens, and the liver was collected and temporarily stored in liquid nitrogen. After that, the serum and liver were stored at −80 °C. Save 1 mm^3^ of liver mass with 4% paraformaldehyde for making pathological sections.

### 2.2. Determination of Production Performance

The feed intake of each replicate was recorded and used to calculate the daily feed intake of each layer. The number of eggs laid was recorded and used to calculate egg production. The feed conversion rate is the ratio of feed intake to average egg weight.

### 2.3. Determination of Egg Quality

There were 8 replicates in each group, and 6 eggs per replicate were selected for egg quality analysis. Egg weight, Haugh unit, albumen height, and yolk colour were measured using a multifunctional egg quality tester (Beijing Brad Technology Development Co., Ltd., Beijing, China). Eggshell strength was determined using an eggshell strength tester (Shandong Shengtai Instrument Co., Ltd., Jinan, China).

### 2.4. Liver Histology

Six replicates were randomly selected from each group, and one liver was selected from each replicate for liver pathological analysis. After the liver tissue block was removed from the 4% formalin solution, it was embedded in paraffin. Subsequently, the paraffin block was cut into sections and stained with hematoxylin and eosin (H & E). The sections were observed and photographed using an upright microscope (Olympus Co., Ltd., Tokyo, Japan). The degree of steatosis was evaluated based on the size and number of vacuoles in the liver.

### 2.5. Determination of Antioxidant Enzyme Activity

The contents of MDA, catalase (CAT), and T-SOD in serum and liver samples were determined using the corresponding commercial kits (Nanjing Jiancheng Biotechnology Co., Ltd., Nanjing, China). The GPX kit was purchased from Solarbio Biotechnology Co., Ltd., Beijing, China. The relevant information about the antioxidant kit is shown in Appendix A.

### 2.6. Transcriptome Analysis

From the results of liver histological sections and antioxidant indicators, the effect of the high-dose group was more obvious, so we only selected the liver of the high-dose group for RNA-seq to explore the mechanism of the two additives. RNA-Seq analyses were performed on the livers of the control group as well as the experimental groups with MSM additions of 700 mg/kg and Se-Y additions of 50 mg/kg. The RNA-seq methods are consistent with Wang et al. (2023) [16]. The quality control results of RNA-seq are shown in Table 2. After sequencing, the differential expression of genes was analysed via DESeq. Genes with an expression fold change (FC) ≥ 1.5 and *p* < 0.05 were selected as differentially expressed genes. After that, a kyoto encyclopedia of genes and genomes (KEGG) enrichment analysis was performed on the differentially expressed genes, and those closely related to this study were listed separately.

### 2.7. Real-Time q-PCR Analysis

To evaluate the accuracy of the RNA-seq results, eight genes, ATP synthase, H^+^ transporting, mitochondrial F1 complex, O subunit (*ATP5O*), ATP synthase, H^+^ transporting, mitochondrial Fo complex subunit G (*ATP5L*), cytochrome c oxidase subunit 8A (*COX8A*), cytochrome c, somatic (*CYCS*), mitogen-activated protein kinase 10 (*MAPK10*), SRC proto-oncogene, non-receptor tyrosine kinase (*SRC*), bone morphogenetic protein 2 (*BMP2*), and fibroblast growth factor 9 (*FGF9*), were randomly selected for real-time q-PCR analysis. The methods of total RNA extraction and cDNA synthesis were the same as those reported by Wang et al. (2023) [16]. Real-time q-PCR was completed on a real-time fluorescence quantitative PCR instrument (Applied Biosystems, Massachusetts, USA), following the manufacturer’s recommendations. The 2^−ddCt^ method was used for quantification with β-actin as a reference. All primers are listed in Table 3.

### 2.8. Statistical Analysis

The software package IBM SPSS statistics 25 (IBM Corp., USA) was used for data analysis, and one-way analysis of variance (ANOVA) was performed for multiple comparisons. The data are expressed as the mean ± standard deviation (SD). Bonferroni’s *t*-test was employed to compare multiple means, and the significance level was set at *p* < 0.05.

## 3. Results

### 3.1. Effects of MSM and Se-Y on Production Performance and Egg Quality

Table 4 shows the changes in production performance and egg quality in the fourth week. Both MSM and Se-Y had no significant effect on the production performance or egg quality of laying hens. With the gradual increase in MSM and Se-Y doses, eggshell strength increased.

### 3.2. Results of Liver Histological Analysis

The results of liver H&E staining are shown in Figure 1. The livers of laying hens in the control group showed significant vacuolization caused by steatosis. With increasing MSM and Se-Y doses, this vacuolization symptom was relieved.

### 3.3. Effects of MSM and Se-Y on Serum and Liver Antioxidant Index Values of Laying Hens

Figure 2 shows the changes in the serum and liver antioxidant index values of laying hens. With the gradual increase in MSM and Se-Y doses, the contents of MDA in serum and liver decreased significantly (*p* < 0.05), which was not the case for the CAT level. The content of T-SOD increased significantly with increasing dosages of both additives (*p* < 0.05). However, neither MSM nor Se-Y had a significant effect on the content of GPX in serum, whereas the GPX level in the liver increased significantly with increasing doses (*p* < 0.05).

### 3.4. Sequencing, De Novo Assembly and Annotation Analysis

A total of 15 libraries were obtained from the three groups (control group, 700 mg/kg MSM, and 50 mg/kg Se-Y), in which 150 bp-paired ends of 50,667,160 to 70,086,200 raw reads were collected, respectively, with Q30 values ranging from 94.02% to 95.15% (Table 1). After eliminating the low-quality raw readings (reads in which the number of bases with a quality score Q < 10 accounted for more than 20% of the total read length), the clean readings obtained ranged from 50,160,676 to 69,126,992. When comparing the clean reading with the *Gallus gallus* genome (Ensembl database), the percentage of total mapped reads ranged from 89.96% to 95.64% (Table 1). The pre-processing results met the requirements for in silico sequencing [17].

### 3.5. Differential Expression and Functional Analysis of Genes

Figure 3 shows the RNA-seq results. When MSM was added at 700 mg/kg, a total of 1116 genes were differentially expressed, of which 612 were upregulated and 503 were downregulated (Figure 3A, Appendix A). When Se-Y was added at 50 mg/kg, 1672 genes were differentially expressed, of which 1202 were upregulated and 470 were downregulated (Figure 3A, Appendix A). Co-expression analysis of differentially expressed genes revealed that a total of 379 genes were co-expressed (Figure 3B). The KEGG pathway enrichment analysis of differentially expressed genes showed that when MSM was added at 700 mg/kg, the pathways with high enrichment and related to hepatic steatosis were “oxidative phosphorylation,” “non-alcoholic fatty liver disease,” and “fat digestion and absorption.” When Se-Y was added at 50 mg/kg, the pathways with high enrichment and related to hepatic steatosis were “inflammatory mediator regulation of transient receptor potential (TRP) channels,” “chemical carcinogenesis,” “pathways in cancer,” and “proteoglycans in cancer” (Figure 3C). The real-time q-PCR results showed that, compared with the control group, the mRNA expression levels of *ATP5O*, *ATP5L*, *COX8A*, and *CYCS* genes in the liver were significantly decreased after adding 700 mg/kg MSM (*p* < 0.05; Figure 3D). After adding 50 mg/kg Se-Y, the mRNA expression levels of *MAPK10*, *SRC*, *BMP2*, and *FGF9* genes in the liver were significantly decreased (*p* < 0.05; Figure 3D). The results of real-time q-PCR were similar to those of RNA-seq, indicating their reliability.

### 3.6. Differential Genes in the KEGG Pathway

The transcriptome responses of layer liver to MSM and Se-Y have not yet been reported. According to the results of the KEGG enrichment analysis, Table 5 summarises the differentially expressed genes in the liver of laying hens after treatment with MSM and Se-Y and lists the Log2 FC multiple of each gene.

## 4. Discussion

With the continuous improvement of breeding levels and feeding technology, the elimination cycle of laying hens can be extended to approximately 72 weeks [18]. Due to the high intensity of egg production and lack of exercise, the incidence of FLS in laying hens during the late peak period of egg production is extremely high, resulting in serious economic losses [1,19]. This study found that adding MSM and Se-Y to the diet of laying hens had no significant effect on production performance or egg quality, which is consistent with the results reported by Kim et al. (2022) and Han et al. (2017) [7,20]. In addition, some studies reported that 0.1% MSM has adverse effects on the feed intake of laying hens [21], which is different from the results of this study. The reason may be related to the different breeds, ages, and experimental cycles of laying hens.

The liver of laying hens needs to synthesise a large amount of lipids for the formation of egg yolk, and laying hens are therefore prone to produce FLS [1,3]. Based on the results of this study, both MSM and Se-Y can alleviate the symptoms of FLS caused by lipid metabolism disorders. Although there is no report on the effects of MSM and Se-Y on liver steatosis in laying hens, lipid oxidation and deposition in the liver are important factors in the formation of liver steatosis [22,23]. Zhang et al. (2023) reported that MSM could reduce lipid oxidation in plasma and pectoral muscle [24]. Maurice (1979) found that Se-Y could reduce the fat deposition in the liver of laying hens and reduce liver bleeding [25]. Therefore, both MSM and Se-Y can protect the liver of laying hens and alleviate the damage caused by abnormal lipid metabolism, which is similar to the results of this study.

Oxidative stress refers to the increase in intracellular levels of oxygen free radicals, leading to oxidative damage to lipids [26,27]. The MDA is the most direct index reflecting the degree of lipid peroxidation [26,27]. Kim et al. (2022) reported that MSM could significantly reduce the content of MDA in the liver of laying hens [7]. In addition, Han et al. (2017) showed that adding 0.3 mg/kg sodium selenite and Se-Y to the diet would lead to a decrease in MDA content in the liver of laying hens [20]. The results of this study showed that both MSM and Se-Y could significantly reduce the content of MDA in serum and liver, which was similar to the reported results, indicating that both MSM and Se-Y can alleviate the degree of lipid peroxidation in layers. The factors CAT, T-SOD, and GPX reflect the antioxidant capacity of the body. Kim et al. (2022) reported that 2 g/kg MSM can significantly increase the content of T-SOD in serum, and the contents of GPX and CAT in the liver and serum of laying hens showed an upward trend [7]. In addition, Han et al. (2017) reported that adding Se-Y to the diet significantly increased the T-SOD content in the liver of laying hens, whereas the GPX content in serum and liver showed an upward trend [20]. The results of this study are similar to those reported previously, indicating that both MSM and Se-Y can improve the antioxidant capacity of laying hens.

The main function of mitochondria is to produce ATP to provide energy for the body, which is also accompanied by the generation of large amounts of reactive oxygen species [28]. When reactive oxygen species accumulate in large quantities in the liver, lipid oxidation damage occurs [4,28]. Genes such as *ATP5I*, *AYP5G1*, *ATP5O*, and *NDUFS1* are related to mitochondrial energy metabolism [29,30]. The effect of MSM on mitochondrial energy metabolism has not been reported yet. However, Sun et al. (2021) showed that knocking down the *ATP5I* and *ATP5G1* genes in human colorectal cancer cells significantly reduced the levels of reactive oxygen species and ATP in the cells [29]. Packialakshmi et al. (2022) reported that after the downregulation of *ATP5O* and *NDUFS1* genes in the renal cortex, the biological function of mitochondria was inhibited [30]. In this study, the addition of MSM to the diet could significantly reduce the expression of genes related to mitochondrial energy metabolism in the liver. When the energy metabolism of the liver mitochondria is inhibited, the content of reactive oxygen species produced by mitochondria decreases, and the oxidative damage to the liver is also alleviated [26]. From the results of this study, the inhibition of mitochondrial energy metabolism may be an important reason why MSM can alleviate liver lipid degeneration damage in laying hens.

Based on the results obtained from the liver histological sections, MSM can alleviate liver steatosis in laying hens; steatosis is an important pathological feature of non-alcoholic fatty liver [4]. This study found that MSM could significantly downregulate the expression of *CYCS*, *NDUFA9*, and *UQCRQ* genes in the liver. Of these, CYCS is the core component of the electron transport chain in mitochondria [31], and the *CYCS* gene is associated with liver lipid metabolism [32]. NDUFA9 is a subunit of the mitochondrial membrane respiratory chain NADH dehydrogenase (Complex I). Zeng et al. (2021) reported that NDUFA9 may participate in the formation of non-alcoholic fatty liver by regulating the activity of the NADH dehydrogenase complex [33]. Canto et al. (2012) showed that the activation of NAD (+) in mammals leads to an increase in the body’s oxidative metabolism, which is an important cause of non-alcoholic fatty liver disease [34]. The UQCRQ is related to the oxidative phosphorylation pathway; oxidative phosphorylation is closely related to non-alcoholic fatty liver disease, and Zeng et al. (2021) assumed that the *UQCRQ* may be a key gene in the pathogenesis of non-alcoholic fatty liver disease [33]. In this study, MSM could downregulate the expression of these three genes, indicating that MSM alleviates non-alcoholic fatty liver disease.

Se-Y has anti-inflammatory effects, and inflammation is closely related to FLS [12,35]. In this study, Se-Y could significantly downregulate the expression of liver genes such as *MAPK10*, *MAPK11*, and *SRC*. The MAPK is mainly activated in response to inflammatory cytokines and cellular stress (including oxidative stress), and MAPK10 and MAPK11 are important members of the MAPK pathway [36,37]. It has been reported that MAPK10 can regulate the inflammatory response in human osteoarthritis and astrocytes [36], and MAPK11 can regulate the inflammatory response in human hepatoma cells and mice [37]. The SRC is involved in the inflammatory regulation of the body [38,39]. Ren et al. (2019) reported that flavonoid urushigenin can inhibit the MAPK signalling pathway mediated by SRC to alleviate inflammation [38], and Byeon et al. (2012) reported that SRC can participate in regulating macrophage-mediated inflammatory responses [39]. In the present study, Se-Y downregulated the expression of these three genes in the liver, indicating that Se-Y can regulate the liver’s inflammatory response.

Se-Y is an antioxidant with cancer prevention effects [40], and *BMP2*, *FGF9*, and *HEY1* are a class of genes associated with the cancer pathway [41,42,43,44,45]. BMP2 is associated with the occurrence of various cancers and can inhibit the proliferation of gastric cancer cells by downregulating EZH2 [41]. Fukuda et al. (2021) reported that BMP2 enhances the proliferation of ovarian and endometrial cancer cells through c-KIT induction [42]. The abnormal activation of FGF9 is associated with many cancers. Chang et al. (2021) reported that FGF9 can promote lung cancer cell proliferation and liver metastasis [43], and according to Wang et al. (2019), it is involved in the migration and invasion of gastric cancer cells [44]. HEY1 is associated with the occurrence of various tumours, and inhibition of HEY1 inhibits the proliferation and migration of liver cancer cells [45]. In this study, Se-Y significantly downregulated the expression of *BMP2*, FGF9, and *HEY1* in the liver, indicating that it can prevent liver cancer in laying hens.

There was no significant difference between the hepatic inflammatory pathway of MSM and the mitochondrial energy metabolism pathway of Se-Y; therefore, we did not discuss it in this study. Based on the results of this study, although the mechanisms of action of MSM and Se-Y are different, they can both alleviate FLS in laying hens, indicating that antioxidant and anti-inflammatory pathways are two important ways to alleviate FLS. Whether MSM and Se-Y have combined effects requires further research.

## 5. Conclusions

Both 700 mg/kg MSM and 50 mg/kg Se-Y can alleviate FLS in 55-week-old layers at the late peak laying period and alleviate the symptoms of steatosis. The mechanism of action of MSM is to increase the level of antioxidants and downregulate the energy metabolism of mitochondria and the expression of genes related to non-alcoholic fatty liver. The mechanism of action of Se-Y is to increase the antioxidant level of laying hens and downregulate the expression of genes related to inflammation and cancer pathways. This study provides a reference for reducing the FLS in laying hens during the late peak laying period.

## Figures and Tables

**Figure 1 animals-13-02466-f001:**
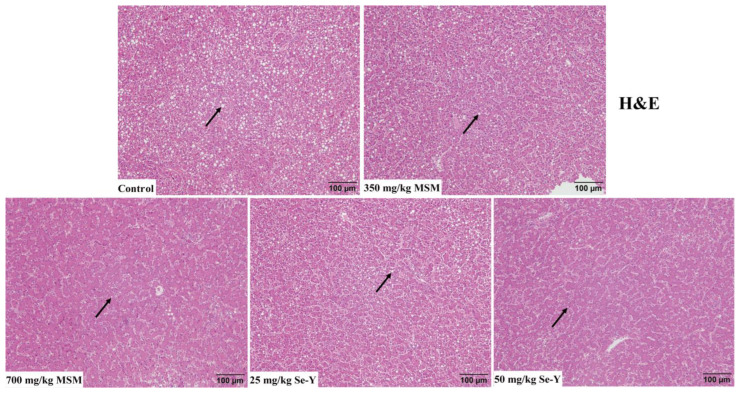
Effects of MSM and Se-Y on liver tissue of laying hens (×200). The arrow indicates the vacuolation symptoms of hepatic steatosis.

**Figure 2 animals-13-02466-f002:**
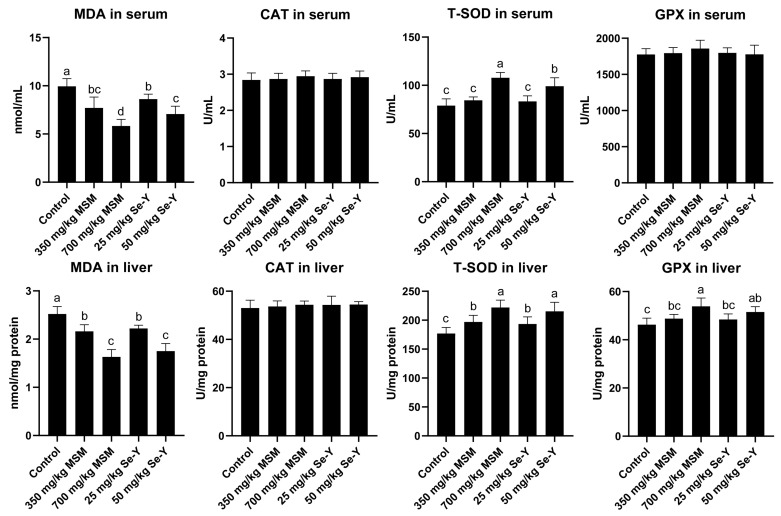
Effects of MSM and Se-Y on antioxidant parameters in the serum and liver of laying hens (*n* = 6). ^a,b,c,d^ Different lowercase letters indicate significant differences between the groups (*p* < 0.05).

**Figure 3 animals-13-02466-f003:**
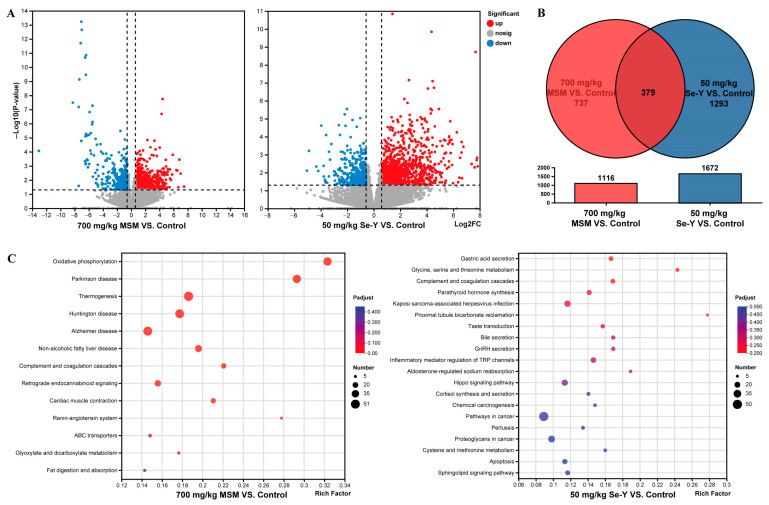
RNA-seq results and analysis. (**A**) Volcanic diagram of differentially expressed genes (red means up, blue means down); (**B**) Wayne diagram of differentially expressed gene co-expression; (**C**) KEGG pathway enrichment analysis of differentially expressed genes; (**D**) Verifying the accuracy of RNA-seq through q-PCR (*n* = 6). ^a,b^ Different lowercase letters indicate significant differences between the groups (*p* < 0.05).

**Table 1 animals-13-02466-t001:** Composition of basal diets and nutrient levels.

Ingredients	Percentage (%)	Nutrient Level	Content
Wheat	68.40	ME (kcal/kg)	2750
Soybean meal ^1^	15.00	Crude protein (%)	16.50
Soybean oil	1.30	Calcium (%)	3.60
DDGS ^2^	3.00	Available phosphorus (%)	0.35
Limestone	8.70	Ether extract (%)	5.41
CaHPO_4_	0.80	Ash (%)	12.32
Bran ^3^	1.00	Met (%)	0.35
Salt	0.30	Lys (%)	0.80
Premix ^4^	1.50	Met + Cys (%)	0.70

^1^ Soybean meal, Protein content 43.7%. ^2^ DDGS, distillers dried grains with solubles, a by-product of corn. ^3^ Bran, a by-product of Wheat. ^4^ The premix provided the following per kg of diet: vitamin A, 15,000 IU; cholecalciferol, 4000 IU; vitamin E, 50 mg; thiamine, 3.60 mg; riboflavin, 10 mg; pyridoxine, 5.5 mg; iron, 100 mg; manganese, 110 mg; copper, 10 mg; zinc, 100 mg; selenium, 0.4 mg; iodine, 0.3 mg.

**Table 2 animals-13-02466-t002:** Statistical summary of the liver RNA-seq datasets.

Sample ^1^	Raw Reads Number	Q30 Value ^2^	Clean Reads Number	Total Mapped Reads Percentage
Control-1	51,129,902	95.15	50,728,872	95.32%
Control-2	55,289,378	94.80	54,763,456	94.28%
Control-3	61,373,938	94.89	60,818,000	94.13%
700-MSM-1	52,676,984	95.12	52,241,862	95.64%
700-MSM-2	55,425,860	94.67	54,862,322	92.94%
700-MSM-3	58,473,686	94.57	57,767,186	92.06%
700-MSM-4	70,086,200	94.19	69,126,992	91.89%
700-MSM-5	55,554,098	94.44	54,884,866	92.97%
700-MSM-6	54,657,056	94.02	53,737,510	89.96%
50-Se-Y-1	52,359,938	94.05	51,544,302	91.58%
50-Se-Y-2	53,882,128	94.65	53,327,412	93.85%
50-Se-Y-3	52,965,636	94.83	52,490,900	94.14%
50-Se-Y-4	59,031,608	94.70	58,413,278	93.38%
50-Se-Y-5	56,635,628	94.30	55,895,582	91.86%
50-Se-Y-6	50,667,160	94.76	50,160,676	93.57%

^1^ 700-MSM means the liver samples from the diets supplemented with MSM at doses of 700 mg/kg, 50-Se-Y means the liver samples from the diets supplemented with Se-Y at doses of 50 mg/kg. ^2^ Q30 values mean the sequencing quality values that correspond to a 0.1% chance of error.

**Table 3 animals-13-02466-t003:** Primers for PCR ^1^.

Gene Name	Accession	Primer Sequence (5′-3′)	Product Size (bp)
*ATP5O*	XM_416717	F: CCTGCTTGCTGAGAATGGTC	210
		R: GAGGGATCGGTCTTGGTCTC	
*ATP5L*	XM_015298211	F: TGGTACTACGCTAAGGTCGAG	324
		R: GCCTCGTTTGCCTATGATCTC	
*COX8A*	XM_001235548	F: AACCAGTGGCAGAGCGATAT	295
		R: CCGCTTCTTGTAGTCCTCGA	
*CYCS*	NM_001079478	F: CCAGAAATGTTCCCAGTGCC	318
		R: AGACTTCTTCTTGATACCCGCA	
*MAPK10*	NM_001318224	F: CTGGTGATGGAGCTGATGGA	294
		R: CTTGTAGCCCATTCCCAGGA	
*SRC*	NM_205457	F: CTGCTTTGGAGAGGTCTGGA	242
		R: ACTTGCCCATCTCTCCCTTC	
*BMP2*	NM_001398170	F: CAACAGCAGCTACCATCACC	206
		R: GAACCACCTCCACCACAAAC	
*FGF9*	NM_001397365	F: AGACAGCGGACTCTACCTTG	253
		R: AGGGTCCACTGGTCTAGGTA	
*β-actin*	NM_205518.2	F: AGTACCCCATTGAACACGGT	197
		R: ATACATGGCTGGGGTGTTGA	

^1^ Abbreviation represents: *ATP5O*, ATP synthase, H+ transporting, mitochondrial F1 complex, O subunit; *ATP5L*, ATP synthase, H+ transporting, mitochondrial Fo complex subunit G; *COX8A*, cytochrome c oxidase subunit 8A; *CYCS*, cytochrome c, somatic; *MAPK10*, mitogen-activated protein kinase 10; *SRC*, SRC proto-oncogene, non-receptor tyrosine kinase; *BMP2*, bone morphogenetic protein 2; *FGF9*, fibroblast growth factor 9; *β-actin*, actin beta.

**Table 4 animals-13-02466-t004:** Effects of MSM and Se-Y on production performance and egg quality ^1^.

Treatment	Control	25 mg/kg Se-Y	50 mg/kg Se-Y	350 mg/kg MSM	700 mg/kg MSM
Feed intake (g/bird)	113.27 ± 2.70	112.45 ± 3.22	113.12 ± 1.86	112.61 ± 2.53	112.84 ± 3.15
Egg production (%)	85.80 ± 5.37	84.93 ± 4.62	85.27 ± 3.65	85.33± 3.79	84.65 ± 5.83
Feed conversion ratio	1.92 ± 0.05	1.88 ± 0.04	1.91 ± 0.04	1.91 ± 0.05	1.90 ± 0.04
Egg weight (g)	59.46 ± 3.61	60.27 ± 5.14	59.61 ± 4.35	59.32 ± 4.15	59.55 ± 5.37
Eggshell strength (N)	36.00 ± 6.50	37.11± 5.86	38.15 ± 6.43	37.62 ± 6.13	37.84 ± 6.25
Haugh unit	81.91 ± 7.13	83.09 ± 7.32	82.47 ± 6.55	81.94 ± 7.36	83.27 ± 6.33
Albumen height	6.86 ± 1.01	7.44 ± 1.25	7.26 ± 1.03	7.14 ± 1.12	6.91 ± 1.11
Yolk colour	13.00 ± 1.35	12.57 ± 1.16	13.58 ± 1.42	12.89 ± 1.35	13.24 ± 1.22

^1^ Values are expressed as means ± SD (*n* = 8).

**Table 5 animals-13-02466-t005:** Differential genes in enriched pathways in the livers of laying hens.

Gene ID	Gene Symbol	Log2 FC	*p*-Value	Gene Description
700 mg/kg MSM VS. Control				
Mitochondrial energy metabolism				
769146	*ATP5I*	−0.73	0.030	ATP synthase, H^+^ transporting, mitochondrial Fo complex subunit E
419992	*ATP5G1*	−0.69	0.007	ATP synthase, H^+^ transporting, mitochondrial Fo complex subunit C1 (subunit 9)
418508	*ATP5O*	−0.59	0.025	ATP synthase, H^+^ transporting, mitochondrial F1 complex, O subunit
418477	*ATP5J*	−0.60	0.015	ATP synthase, H^+^ transporting, mitochondrial Fo complex subunit F6
101749042	*ATP5L*	−0.72	0.009	ATP synthase, H^+^ transporting, mitochondrial Fo complex subunit G
770190	*COX17*	−0.90	1.36 × 10^−5^	COX17, cytochrome c oxidase copper chaperone
420243	*COX6C*	−0.79	0.022	cytochrome c oxidase subunit 6C
772260	*COX7A2*	−0.71	0.013	cytochrome c oxidase subunit 7A2
431629	*COX7C*	−0.97	0.001	cytochrome c oxidase subunit 7C
775974	*COX8A*	−0.82	0.035	cytochrome c oxidase subunit 8A
771510	*NDUFS1*	−0.61	0.032	NADH:ubiquinone oxidoreductase subunit S1
63549497	*ND6*	−0.87	0.014	NADH dehydrogenase subunit 6
Nonalcoholic fatty liver disease				
420624	*CYCS*	−0.68	0.036	cytochrome c, somatic
768860	*NDUFA9*	−0.82	0.015	NADH:ubiquinone oxidoreductase subunit A2
424078	*NDUFB3*	−0.64	0.026	NADH:ubiquinone oxidoreductase subunit B3
404751	*NDUFC2*	−0.65	0.017	NADH dehydrogenase (ubiquinone) 1, subcomplex unknown, 2
416336	*UQCRQ*	−0.68	0.025	ubiquinol-cytochrome c reductase complex III subunit VII
50 mg/kg Se-Y VS. Control				
Hepatic inflammatory pathway				
422592	*MAPK10*	−1.38	0.042	mitogen-activated protein kinase 10
417739	*MAPK11*	−0.91	0.033	mitogen-activated protein kinase 11
396442	*SRC*	−0.62	0.044	SRC proto-oncogene, non-receptor tyrosine kinase
418212	*ITPR2*	−2.54	0.032	inositol 1,4,5-trisphosphate receptor type 2
419910	*ITPR3*	−2.38	0.008	inositol 1,4,5-trisphosphate receptor type 3
Hepatic cancer pathway				
378779	*BMP2*	−0.91	0.014	bone morphogenetic protein 2
419311	*CTSZ*	−0.78	0.040	cathepsin Z
378917	*FGF9*	−0.74	0.045	fibroblast growth factor 9
428365	*HEY1*	−1.09	0.025	hes related family bHLH transcription factor with YRPW motif 1
427339	*LPAR1*	−1.21	0.025	lysophosphatidic acid receptor 1
395561	*WNT4*	−1.07	0.045	wnt family member 4

## Data Availability

Raw data are held by the author and may be available upon reasonable request.

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
