# Peer review of "Effects of Methyl Sulfonyl Methane and Selenium Yeast on Fatty Liver Syndrome in Laying Hens and Their Biological Mechanisms"

_animals, 2023, doi:10.3390/ani13152466_

Round 1
Reviewer 1 Report
Dear authors,
in your manuscript you presented the effect of MSM and Se-Y on the FLS in laying hens. The paper is written in good English, Materials and methods are described in such a way that they allow repeatability, and the Results are clarified in the Discussion.
Specific comments:
L 44 - what do you mean by "Slow liver lesion"?
L 50 - "improve" oxidative stress - maybe to use another word instead of improve?
L 64-67 - please revise ( poultry production -production performance of poultry...)
L 68-74 to many "laying hens"
L 92 "eggs of the fourth week" - please elaborate did you collect the eggs only in the weak 4 of the experiment or you mean that you collected the eggs during the 4-week experiment?
Reviewer 2 Report
To improve the quality of work, I would like to see the following information:
1. Add the name of the used equipment in the study of liver histology.
2. To expand the description of the methods that determine the antioxidant status and the equipment used.
3. Check the design of the source 31 In the references.
Reviewer 3 Report
e authors described a study on Methyl sulfonyl methane and selenium yeast in alleviating fatty liver in laying hens. This article explored the possibility of Methyl sulfonyl methane and selenium yeast to alleviate fatty liver from two aspects of antioxidant and anti-inflammatory. In addition, this study also explored the Transcriptome reaction of layer liver to Methyl sulfonyl methane and selenium yeast for the first time, which is a good supplement to the existing research on layer fatty liver. The results of this study are relevant, and due to the above reasons, after minor revisions, it can be accepted for publication in the Journal of《Animals》.
1. Why did we only do RNA-seq of 700 mg/kg Methyl sulfonyl methane and 50 mg/kg selenium yeast? What about the low-dose group?
2. Although the authors discuss about several genes based on the results of RNA-seq, they should verify the expression of the genes by real-time PCR like Fig 3D.
3. In Table 1, the full name of DDGS should be spelled out.
4. In the Table 2, the first letter of each header should be capitalized, and the following should be changed to lowercase, such as Modify "Raw Reads Number" to "Raw reads number".
5. Lines 128 and lines 157: Please unify lowercase “P” and change it to “p”. Please check other parts of the article.
6. Lines 173: Please mark the pathological changes on Figure 1.
Reviewer 4 Report
The article "Effects of Methyl Sulfonyl Methane and Selenium Yeast on Fatty Liver Syndrome in Laying Hens and their Biological Mechanisms" represents short-term data retrieved after 4 weeks of feeding experimental diets and presents only one week of production data. The authors hypothesise that MSM and SeY products can reduce FLS in mature hens (59 weeks old) via oxidative phosphorylation and anti-inflammatory effects, respectively. The article lacks some fundamental information to allow the reader to understand the study design. The article needs to be revised and resubmitted to be considered for publication. Addressing the following points may help to improve the manuscript.
|
Line number |
Reviewer comments |
|
14-15 |
Delete sentence. Your data shows that higher doses of MSM and SY may help reduce lipid oxidation and inflammation and, thus, eventually reduces the FLS in 390-day-old hens. However, it is not clear which marker indicates prevention. The data is not exclusive to provide a reference. The conclusion is misleading. Stick to facts. |
|
24 |
T-SOD and GPX should be used as full words followed by abbreviations in brackets |
|
28 |
Suggest the term "reduce" to replace "improve". |
|
50 |
The term "studies" means more than one study. Add more references |
|
Abbreviations |
MDA, MAPK, CAT, T-SOD, GPX , KEGG, TRP and many other terms are used throughout the article. All these should be used as full words, followed by abbreviations in brackets. |
|
Material and Methods section |
This section needs to be revised to include the following information: 1. Why this age of hens was selected for this study? 2. The age of hens should be in weeks 3. Where were birds reared? Add housing details? 4. What were birds fed before the experimental period? 5. How were MSM and Se-Y added to the diets? Added on the top of the formulation or part of the formulation? Was the test product liquid or powder? How was it mixed? 6. Mention the form of the diet. Mash or pellets 7. Line 91-92:The experimental period was 4 weeks, but why was only one week of egg production recorded? Why not 4 weeks? 8. Line 93: How many hens/replicate were used for blood- and liver analysis? Mention the number for each analysis. 9. Mention body weight and weight gain for the experimental period. 10. How were liver samples kept before being frozen at -80C? |
|
Table 1 |
1. Mention the protein percentage of soybean meal used in brackets. 2. DDGS: mention the byproduct of which cereal? Coen, wheat or sorghum? 3. Bran: Which bran? 4. Mention salt level? 5. Was any yellow colour pigmentation added to the diet? 6. In the nutrient level section, add data on crude fat, sodium, ash, vitamin E or any other antioxidant added. 7. Why was MSM only tested for Mitochondrial energy metabolism and Se-Y for inflammatory pathways? Why not for both? Give reason. 8. In the introduction section, mention if the hepatic cancer pathway genes are relevant in healthy laying hens. I have never heard of cancer in hens. Are the authors implying that FLS can lead to cancer in humans or hens? The connection to these analyses is not clear. Why hepatic cancer pathways were tested is not clear. |
|
Section 2.2 |
Was feed intake recorded for each week or for 4th week only? Clarify and mention? |
|
Section 2.3 |
How many eggs /replicate were used for egg quality analysis? mention |
|
Section2.4: |
how many samples/replicate? Add information on how photographs were taken and how the vacuolisation was scored. Add reference. |
|
Section 2.5 |
Add a table to mention which takes were used and add a source and reference. |
|
Section 2.6 |
Why only higher doses of MSM and Se-Y were tested? Given the reasoning in the methods and material section. |
|
Table 4 |
n = 8 for production data means only one egg or hen was used. How is it possible? The authors need to clarify. The authors may want to say that n=8 replicates/treatment for production parameters and 1 egg/replicate for egg quality. Check and revise. |
|
Table 5 |
Add a column to add the P-value for each gene tested. |
|
Section 3.4 |
Mention the criteria used to eliminate the low-quality raw readings. |
|
Line 236-237 |
This is a big statement based on only one week's data. Not sure why weekly data was not collected. |
|
Line 239 |
What was the level of MSM used in those studies where adverse effects were recorded? Mention levels. |
|
Line 236-237 |
What was the duration of the study reported by Kim et al. |
|
Line 244 |
Be consistent with the use of terminology. The term “hepatic steatosis” or FLS ? Stick to one term. |
|
Discussion in general |
If the mode of action of MSM is via the reduction in lipid oxidation, then maybe an increase in Vitamin E or any synthetic antioxidant in the diet can help. Why use MSM? Was any antioxidant used as a part of the premix in these experimental diets? Clarify these points in the discussion. What is in MSM that can cause a reduction in lipid oxidation? Explain. |
|
Line 255-256 |
What was the level of Se used in Han et al diet? Add details. Also, mention how SeY was given in the diet (source). Was it the same source used in the current experiment? |
|
Line 314-326 |
Not sure what the authors are trying to prove here. Are these genes relevant in poultry? Revise this paragraph to clarify the message. Are authors implementing that FLS can cause cancer if not treated? What is meant by the abnormal activation of FGF9 or HEY1? Was it abnormal in this study? At what inclusion level can it prevent liver cancer? Is this study a suitable model for studying cancer cells? Please justify. |
|
Line 327-320 |
The authors have only tested one mode of action for each product. Why? It needs to be clarified that there may be other modes of action, but in this study, only one mode was tested. Maybe the mode of action for both is the same. You have not tested, so you can't confirm that they work differently? Revise the sentence to clarify this. |
|
Conclusion |
Rewrite conclusion. Mention the age of the hens and mention what happens at what levels. Stick to the study outcomes? Are you suggesting that MSM at 700mg/ and Se-Y at 50mg/kg be used in 55 to 59-week-old hens? Not clear. What is the practical implication of your study? Not clear? Delete- "This study provides a reference and theoretical basis for preventing FLS in laying hens during the late peak laying period".What do you mean by theoretical basis? And the data set is too small to provide a reference. |
Round 2
Reviewer 4 Report
The manuscript ID Animals 2499368V3 has been considerably improved. However, to be accepted for publication, it requires further minor revisions. Comments below may help to improve the manuscript:
Line 86: Table 1 has a reference. If you have used the same formulation as in reference 15, then there is no need to provide a table on feed formulation. Refer to it. Please check and amend.
Line 87: You have used the term "powdered form"? Does this mean the meal basal diet was ground to bring the texture to powder form? Please check the terminology. Also, mention what the steps were. The statement that the product was added step by step needs further clarification.
Line 98: Be scientific and mention the quantity in grams. "Little liver" is not a scientific way of describing quantities.
Line 274-276: mention the quantity of test product used by Kim et al.
Line 351: Are you suggesting that your recommended doses will prevent FLS? Does your research show that FLS is completely treatable? I suggest using the term" helped reduce FLS." as we can still see vacuolation symptoms in the highest inclusion doses.
In response to the question, "Why was MSM only tested for Mitochondrial energy metabolism and Se-Y for inflammatory pathways? Why not for both? Give reason." Your response was: "These two pathways are obtained by RNA-seq analysis. We sequenced the livers of the two additives and listed the significantly different pathways in the manuscript. There was no significant difference between the inflammatory pathway of MSM and the mitochondrial energy metabolism pathway of Se-Y, so we do not list in the manuscript" Add your response in the result section.
In response to comment 27: for section 2.5, if adding another table is not an option, you can easily add this information by adding the references to kits in paragraph form and a reference for each kit used. This information is mandatory and thus must be added.
